# Effect of Low-Density Lipoprotein (LDL) and High-Density Lipoprotein (HDL) on Frozen Gelation of Egg Yolk

**DOI:** 10.3390/foods14030522

**Published:** 2025-02-06

**Authors:** Junze Yuan, Songyi Lin, Kun Liu, Fujun Guo, Zhijie Bao

**Affiliations:** 1National Engineering Research Center of Seafood, School of Food Science and Technology, Dalian Polytechnic University, Dalian 116034, China; 16604119745@163.com (J.Y.); linsongyi730@163.com (S.L.); 2Engineering Research Center of Special Dietary Food, The Education Department of Liaoning Province, Dalian 116034, China; 3Dalian Green Snow Egg Products Development Co., Ltd., Dalian 116036, China; 13700058961@163.com (K.L.); 15998587970@163.com (F.G.); 4Engineering Research Center of Food, The Education Department of Liaoning Province, Dalian 116034, China

**Keywords:** egg yolk, low-density lipoprotein (LDL), high-density lipoprotein (HDL), freeze gelation

## Abstract

This study aimed to investigate the roles of low-density lipoprotein (LDL) and high-density lipoprotein (HDL) in the gelatinization behavior of egg yolk, as well as the underlying mechanisms of action. This research examined the rheological properties, moisture distribution, and structural characteristics of a system containing reconstituted egg yolk components during the freezing process. The results indicated that increasing the concentration of LDL and HDL in the egg yolk system enhanced the apparent viscosity of egg yolk following a freeze–thaw treatment. Specifically, as the LDL and HDL content increased, *G*’ and *G*” values increased significantly, whereas tanδ values decreased significantly and *l** values declined. These findings suggest that both LDL and HDL are critical contributors to the gelatinization process of egg yolk. Furthermore, as the concentrations of LDL and HDL in the system increased, the amount of fixed water also rose, while the bound and free water content decreased. This observation implies that LDL and HDL facilitate water migration during the freezing of egg yolk. The increase in fluorescence intensity observed in the fluorescence spectra indicates a greater exposure of tyrosine residues on the protein surface, an enhancement of surface hydrophobicity, and a modification of protein conformation. Fluorescence inverted microscopy revealed that elevated levels of LDL and HDL in the system led to increased structural damage to the protein due to freezing, which subsequently promoted the aggregation of yolk proteins. This suggests that both LDL and HDL undergo aggregation during gelation. In egg yolk, LDL and HDL are essential for gel formation during the freezing of liquid egg yolk and play critical roles in both protein structure and water migration. Of the two lipoproteins, HDL has a more pronounced effect on gel formation during liquid egg yolk freezing. This study investigates the key substances involved in the gelatinization of egg yolk, providing a reference for further improvements in egg yolk gelatinization during freezing.

## 1. Introduction

The egg yolk is the most complex component of the egg, comprising approximately 26% to 33% of its total weight. It exhibits excellent emulsifying properties, making it a high-quality food emulsifier commonly utilized in the production of various food products, including mayonnaise and salad dressing. Additionally, liquid egg yolk possesses significant nutritional value and unique functional characteristics, leading to its widespread application across multiple industries, including medicine, pharmaceuticals, cosmetics, and food. In food applications, egg yolks are frequently frozen to enhance transportation and storage efficiency while extending their shelf life. The advantages of freezing egg yolks include the prevention of microbial growth and spoilage, the preservation of flavor and color, and the inhibition of various degradation chemical reactions [1]. However, the gelatinization of egg yolks occurs at temperatures below −6 °C. This gelatinization effect causes iced egg yolks to lose the fluidity characteristic of fresh egg yolks, rendering them undispersible even through mixing. Mechanical treatments may disperse the egg yolk into speckles, which complicates its integration with other food components. Additionally, the thawing time may be extended or may not occur at all, significantly impacting its application in downstream food enterprises [2]. Consequently, there is an urgent need for effective methods to deter the occurrence of frozen egg yolk gelation, necessitating an in-depth exploration of the mechanisms underlying this phenomenon. The gelation of egg yolk limits its application in food products, prompting several scholars, both domestically and internationally, to investigate methods for inhibiting the gelation of frozen egg yolks. Various approaches, including physical, chemical, and enzymatic methods, have been employed to mitigate the effects of gelatinization on the quality of egg yolks. Pearce et al. [3] found that mechanical treatments, such as the homogenization or colloidal milling of egg yolks prior to freezing, partially prevented the increase in the apparent viscosity of frozen egg yolks. Similarly, Burley and Cook [4] discovered that increasing the freezing rate of iced egg yolks decreased the extent of gelatinization. Izat and Gardner [5] reported that the enzymatic treatment of egg yolks with trypsin inhibited yolk cryogelation. In a separate study, Ma et al. [6] investigated frozen egg yolks over a 30-day period and found that the enzyme-free yolks underwent gelatinization, while the enzymatically solubilized yolks exhibited properties comparable to those of fresh egg yolks. Urban et al. [7] found that NaCl, along with sugars such as sucrose, alleviates the gelatinization of iced egg yolks. Current practices to inhibit the gelation of iced egg yolks typically involve the addition of salt, sugar, or corn syrup prior to freezing. However, consumer preferences for reduced salt and sugar may restrict the potential applications of iced egg yolks, highlighting the need to investigate healthier and more effective methods for inhibiting gelatinization. While studies on the gelatinization phenomenon of iced egg yolks have been conducted internationally, they primarily focus on the effects of prolonged freezing and storage times. Consequently, the findings are often localized and do not adequately clarify the underlying causes of the gelatinization phenomenon in iced egg yolk [8]. Therefore, the mechanism of the formation and development of ice yolk gelatinization needs to be further studied.

Researchers have proposed various explanations for the gelation mechanism associated with gel composition [9]. Yolk consists of approximately 50% water and 50% dry matter, which can be further categorized into 77–81% slurry and 19–23% granules. The slurry is composed of 85% low-density lipoprotein (LDL) and 15% yolk protein, while the granules contain 70% high-density lipoprotein (HDL), 16% yolk protein, and 12% LDL [10]. The most common explanation for yolk gelation is the formation of large ice crystals during the freezing process. This results in an increased concentration of yolk components and leads to protein aggregation due to the dehydration of the yolk system [8].

There is ongoing debate regarding the components involved in the gelatinization process. Most studies indicate that low-density lipoprotein (LDL) is the substance responsible for the gelation of frozen egg yolks; however, the mechanism of LDL aggregation remains contentious. It has been proposed that the initial stage of egg yolk gelation following freezing and thawing involves the disruption of the LDL micelle structure. Freezing ‘dehydrates’ the deacylated proteins of LDL (resulting in a decrease in water activity, a_w_), disrupts lipoprotein interactions, and enhances nonpolar protein interactions, ultimately leading to gelation [11]. Jolivet et al. [2] attributed the increase in LDL concentration to the iced egg yolk, which, after the removal of water, causes the aggregation of LDL and subsequently induces gelation. Kiosseoglou and Paraskevopoulou [12] attributed this aggregation to protein–protein interactions that occur following the breaking of the bond between lecithin and protein on LDL. Kurisaki et al. [13] utilized gel chromatography and electron microscope scanning to conclude that freezing causes the release of apolipoproteins and phospholipids from the surface of low-density lipoprotein (LDL). This dissociation of LDL′s combined components leads to the destabilization of its structure, contributing to LDL aggregation. Furthermore, some researchers have indicated that high-density lipoprotein (HDL) plays a significant role in freeze-induced yolk gelatinization. It was observed that the presence of yolk particles enhanced freeze-induced yolk gelatinization compared to yolk slurry alone. A study examining the effects of long-term frozen storage on yolk gelation reported that lipoprotein particle aggregation occurs due to water removal during the initial phase (days 1–28) of slow freezing. In the subsequent phase (days 28–84), stronger gel networks are formed through the release and reaggregation of previously aggregated proteins, either between or within the protein structures [14]. He also suggested that HDL is involved in aggregation, which plays a role in the gelation process. This was demonstrated by probing a variety of metrics, including particle size, matrix mobility, protein aggregation, and microstructure. However, the role of HDL in gelation requires further validation.

This study primarily investigates the changing gelatinization behavior of egg yolk LDL and HDL during freezing. It analyzes the relationship between the structural stability of proteins in LDL and HDL and the gelatinization behavior of egg yolk post-freezing. This research elucidates the rheological properties, water distribution, protein structure, and microstructural changes in LDL and HDL within gelatinized egg yolk after freezing. Furthermore, it aims to identify the key components influencing the egg yolk gelation phenomenon and to reveal the underlying mechanisms of this gelation. This study provides theoretical support for managing quality changes in frozen egg yolks, thereby expanding their applications in the food industry.

## 2. Materials and Methods

### 2.1. Materials

Fresh eggs were procured from Dalian Dahongfu Supermarket and stored in a refrigerator at a temperature of 4 °C for a maximum duration of 7 days. All other chemicals utilized in this study were of analytical grade.

### 2.2. Sample Preparation

Yolks were separated using the method described by Wang et al. [15], with slight modifications. Fresh eggs were beaten by hand, after which the whites and yolks were separated using an egg separator. Each yolk, maintaining an intact yolk membrane, was rolled on a paper towel to remove any residual egg white. The yolk membranes were then broken, and the yolks were collected in a beaker. Subsequently, the yolks were blended in a high-speed blender (JJ-1, Pudong Physical-Optical Instrument Factory, Shanghai, China) for 30 min to ensure thorough mixing. The resulting yolk solution was filtered through a 500 mesh strainer to remove yolk membranes, eggshell residue, and other impurities.

LDL separation was performed according to the method of Ren et al. [16] with slight modifications. The collected yolk liquid was magnetically stirred with an equal mass of NaCl solution (0.17 mol/L) for 1 h. Yolk particles and yolk plasma were separated by centrifugation (Anke, Model TGL-20B, Shanghai, China) at 10,000× *g* for 45 min. The repeated centrifugation of the separated yolk plasma was performed to ensure the complete removal of the yolk particles. The supernatant was then collected, and 5% (wt) polyethylene glycol (PEG4000, Soalrbio, Beijing, China) was added and mixed following 1 h of magnetic stirring. After centrifugation at 15,292× *g* for 15 min, the yellow, oily, paste-like object floating in the upper layer was identified as LDL.

HDL isolation was conducted following the method described by Hollman et al. [17], with minor modifications. The egg yolk solution was magnetically stirred with distilled water in a 1:5 ratio for 1 h to ensure thorough mixing, and subsequently sonicated in an ultrasonic cleaner for 20 min. Centrifugation was carried out using a centrifuge set to 10,000× *g* for 45 min, resulting in the HDL being collected as the bottom precipitate.

The original LDL and HDL contents of egg yolk liquid were 10% and 2.5%, respectively. We subsequently added LDL and HDL to the egg yolk liquid to increase its lipoprotein content by 3%, 6%, and 9% compared to the original levels. However, the addition of HDL and LDL inevitably alters the moisture content of the yolk liquid system. To address this, we utilized a moisture content meter (4700, Hitachi, Kyoto, Japan) to determine the moisture levels of both the yolk liquid and the LDL and HDL. A specific amount of distilled water (DW) was added to the LDL and HDL, which was then thoroughly stirred to match the moisture content of the yolk liquid. This step ensured that the moisture content of the system remained consistent after the addition of LDL and HDL. The egg yolk liquid was then stirred with LDL and HDL in a water bath at 30 °C for 30 min to ensure complete dispersion.

### 2.3. Yolk Freezing and Thawing Treatments

In accordance with various experimental requirements, 20 g of recombinant egg yolk solution at different concentrations were placed into 25 mL culture flasks and frozen at −37 °C for durations of 2, 4, 6, and 8 h. The samples were subsequently thawed by allowing them to stand at 25 °C for 4 h. Fresh egg yolks were also prepared for comparison with the freeze–thawed samples, serving as the control group. Each sample underwent a single freeze–thaw cycle. Frozen egg yolk liquid samples were prepared in at least triplicate, unless otherwise specified.

### 2.4. Observation of Freezing and Thawing Phenomenon of Egg Yolk

Culture flasks containing reconstituted egg yolk solution, after undergoing freezing, were positioned upside down and thawed at room temperature for 4 h. Fresh egg yolks were also prepared for comparison with the freeze–thawed samples, serving as a control. The freeze–thawed egg yolk solution was photographed, and the underlying causes of the observed phenomenon were analyzed.

### 2.5. Rheological Analysis

The rheological properties of the egg yolk mixture system were determined using the method described by Wang et al. [18], with slight modifications. The rheometer (DHR2, Trenton, NJ, USA) was employed to characterize the rheological properties of the samples. An aluminum plate with a diameter of 40 mm and a gap of 1.0 mm was selected as the probe, and the egg yolk samples were tested at a temperature of 25 °C.

#### 2.5.1. Mobility Behavior Analysis

The flow of the samples was determined using the steady-state shear mode of the rheometer. The shear rate was set from 0.1 s^−1^ to 100 s^−1^, and the obtained curves of apparent viscosity versus shear rate were fitted by a power law equation:(1)$$\eta =K\cdot {\dot{\gamma }}^{n}$$
where η denotes the apparent viscosity (Pa⋅s); $\dot{\gamma }$ denotes the shear rate (s^−1^); *K* is the apparent viscosity coefficient (Pa⋅sn); and *n* is the flow index.

#### 2.5.2. Viscoelasticity

The viscoelastic properties of the sample were assessed using the dynamic oscillation mode of the rheometer. Initially, the linear viscoelastic region of the samples was identified in stress mode. Subsequently, in frequency scanning mode, the stress was maintained at 1%, with a frequency range of 0.1 rad/s to 100 rad/s. To mitigate the influence of external forces, all samples were allowed to equilibrate for 1 min prior to testing. The resulting curves for the elastic modulus (G′), viscous modulus (G″), and loss tangent (tan δ) of the samples as a function of frequency were then obtained.

### 2.6. Moisture Distribution (LF-NMR) Analysis

The moisture distribution of the egg yolk mixture was determined using the method described by Au et al. [19], with slight modifications. Measurements of 1H molecular migration and the kinetics of both the frozen egg yolk liquid and frozen egg yolk were conducted using a low-field pulsed nuclear magnetic resonance analyzer (Suzhou Niumag Co., Ltd., Suzhou, China). Egg yolk samples were placed in 25 mL petri dishes within 40 mm diameter test tubes for testing. All samples were examined in four replicates. The CPMG pulse sequence was employed, with the following parameters: waiting time (TW) of 4000 ms, signal superposition (NS) of 8, preamplification (PRG) of 2, echo time (TE) of 0.5 ms, and number of echoes (NECH) set to 8000. The number of iterations for the inversion was established at 10,000, resulting in the acquisition of inverse curves for the samples.

### 2.7. Protein Conformational Change Analysis

#### 2.7.1. Raman Spectrum Analysis

The secondary structure of the egg yolk mixture was assessed through a method adapted from Mahadevan et al. [20]. For the Raman experiments (HORIBA HR Evolution, Horiba Jobin Yvon, France), the egg yolk samples were prepared on slides, with an excitation wavelength set at 532 nm and an emission power of 50 mW. The range of the Raman spectra collected spanned from 400 to 2000 cm^−1^. Each sample underwent at least three scans, and the resulting spectra were averaged to generate a single Raman spectrum for each sample. The peak position error was kept below ±3 cm^−1^. The baseline calibration and analysis of the Raman spectra were performed using OMINIC software. The phenylalanine band at 1003 cm^−1^ was used as the normalized internal standard, enabling the measurement of intensity fluctuations in the other Raman peaks.

#### 2.7.2. Fluorescence Spectroscopy Analysis

The tertiary structure of the egg yolk mixture was determined using a method adapted from Gmach et al. [21]. Structure unfolding studies were conducted based on intrinsic emission fluorescence spectroscopy with a fluorescence spectrophotometer (Hitachi, Kyoto, Japan). The protein concentration was adjusted to 0.1 mg/mL using a 0.01 mol/L sodium phosphate buffer (pH 7.0). The excitation wavelength was set to 280 nm, while the emission wavelengths ranged from 290 to 400 nm. Both the emission and excitation slit widths were maintained at 5 nm, with a scanning speed of 1500 nm/min. The fluorescent moieties within the proteins acted as fluorescent probes, and the phosphate buffer served as the blank.

#### 2.7.3. Hydrophobicity Analysis

The tertiary structure of the mixture containing egg yolk was assessed by employing the method outlined by Liu et al. [22], with minor adjustments. The fluorescent probe used was 1-Aniline-8-naphthalenesulfonate (ANS). To create five distinct mass concentration gradients (0.05, 0.1, 0.2, 0.4 and 0.5 mg/mL, *w*/*v*), egg yolk samples were dissolved and subsequently diluted with phosphate-buffered saline (PBS) at a concentration of 0.01 mol/L and a pH of 7.0. Following this, 4 mL of the diluted samples were combined with 20 μL of ANS solution (8 mmol/L), mixed vigorously for 10 s, and then kept in the dark for 3 min. The measurement of the samples’ fluorescence intensity was performed using a fluorescence spectrophotometer (F-2700, Hitachi, Japan) as a reference. The operating parameters included an excitation wavelength set at 390 nm, an emission wavelength at 470 nm, and a slit width of 5 nm.

### 2.8. Microstructure

The microstructure of the egg yolk mixture was determined using the method of Kocal et al. [23], with slight modifications. The microstructure of the freeze–thawed processed egg yolk liquid was observed through fluorescence inverted microscopy (DM 3000, Leica, Germany). The freeze–thawed egg yolk solution was diluted 10-fold and mixed homogeneously with fluorescein isothiocyanate (FITC) (dissolved in acetone at a concentration of 1 mg/mL) and Nile red (dissolved in ethanol at a concentration of 1 mg/mL) at ratios of 100:1 and 100:3, respectively, in the dark. The stained samples were placed on slides and examined under a 20× objective lens using a forward-inverted integrated fluorescence microscope.

### 2.9. l* Diffusion Wave Spectrometer Analysis

A total of 3.5 g of the egg yolk solution was accurately weighed in a 4 mL cuvette with an optical path length of 10 mm. The cuvette was placed into the sample chamber of the diffusion wave spectrometer (RheoLab III, LS Instruments, Fribourg, Switzerland). ‘Transmission’ was selected for the measurement geometry, the repetition duration for normal measurements was set to 60 s, and for echo measurements, the duration was set to 30 s. Subsequently, the *l** value of the sample was recorded. The measurement geometry was confirmed as ‘Transmission’, the normal repetition duration was set to 60 s, the echo repetition duration was set to 30 s, and the *l** of the sample was recorded.

### 2.10. Statistical Analysis

All data were tested in parallel a minimum of three times. The results were analyzed for significance using SPSS 19.0 software (SPSS Inc., Chicago, IL, USA), with the level of significance set at *p < 0.05* and presented as Mean ± SEM. GraphPad Prism software (version 9.5, San Diego, CA, USA) was utilized for data plotting.

## 3. Results

### 3.1. Freeze–Thawed Egg Yolk Liquid Sample Analysis

As illustrated in Figure 1, we initially froze the yolk solution within the culture flask and subsequently thawed it in an inverted position, allowing the yolk solution to flow downward. After 2 h of freezing, the yolk solution melted and flowed from the top of the culture flask, indicating that its apparent viscosity had not increased sufficiently to form a gel. More yolk liquid from the HDL-added group remained on the walls of the culture flasks relative to the LDL group when thawed and runoff, suggesting that the yolk liquid from the HDL-added group may have a higher apparent viscosity after freezing and thawing. After 4 h of freezing, the yolk liquid in the sample group containing LDL and HDL exhibited greater adherence to the top of the culture flask compared to the control group, suggesting the potential formation of a gel. At a 3% increase in lipoprotein content, a portion of the yolk liquid in the LDL- and HDL-added groups flowed down from the top of the culture flasks, indicating that, at this point, the yolk liquid became more viscous but did not fully gel. When the lipoprotein content was increased to 6%, the yolk liquid in the LDL-added group partially flowed down, whereas the yolk liquid in the HDL-added group remained fixed at the top of the incubator and did not flow down, suggesting that the yolk liquid in the HDL-added group may have formed a gel. Upon increasing the lipoprotein content to 9%, none of the yolk liquid from the LDL and HDL groups flowed down from the top of the culture bottle after freezing and thawing, indicating that the yolk liquid had likely already undergone gelatinization. Therefore, we hypothesized that the addition of HDL would be more likely to result in gel formation after the freezing and thawing of the egg yolk liquid. In contrast, following 6 h of freezing, the yolk liquid had entirely transitioned into a gel state and remained immobilized at the top of the culture flask, showing no further flow. Consequently, we focused on characterizing the sample group that had been frozen for 4 h to explore the mechanisms by which LDL and HDL promote the gelation of the yolk liquid.

### 3.2. Rheological Analysis

#### 3.2.1. Apparent Viscosity Analysis

Rheological properties are closely correlated with both the structural and molecular characteristics of proteins, making them valuable for studying the shear response of gels [15]. This study examined the flow behavior of frozen egg yolk liquid with varying concentrations of sugar alcohols, utilizing a steady-state shear mode rheometer. As illustrated in Figure 2a, the apparent viscosity of the samples after 4 h of freezing changed with the increasing shear rate; all frozen egg yolk liquids exhibited shear-thinning behavior, with apparent viscosity decreasing sharply as the shear rate increased. This phenomenon is likely due to the weakening of intermolecular entanglements. Following freezing, egg yolk proteins aggregate to form a physical gel that is more complex than gels formed solely by intermolecular forces, disrupting the original structure of the egg yolk and resulting in a gelling phenomenon upon thawing [24]. The apparent viscosity of the samples in the LDL and HDL addition groups was greater compared to that of the control group samples and exhibited a gradual increase with higher protein content. This suggests that the increase in LDL and HDL content may facilitate the formation of a frozen gel in egg yolk liquid [25]. The underlying reason for this phenomenon may be attributed to the aggregation of lipoproteins during freezing. The apparent viscosity remains unchanged in the groups with the 6% and 9% HDL addition. This may be attributed to the fact that once the HDL content reaches 6%, the gel network within the yolk is already fully formed, resulting in high gelation. Consequently, further increases in HDL do not lead to an increase in apparent viscosity. Notably, the apparent viscosity of the HDL-added group was higher than that of the LDL-added group, indicating that HDL has a more pronounced effect on the freezing gelation of the egg yolk liquid. This observation aligns with the apparent changes in egg yolk liquid following freezing and thawing. The rationale for this may lie in the fact that HDL molecules are more prone to interacting with proteins to form aggregates, which play a crucial role in gel formation. It has been suggested that HDL contributes to gelation by forming larger aggregates with other protein components than those formed by protein components alone [26].

#### 3.2.2. Modulus of Elasticity and Loss Modulus Analysis

The change in the viscoelastic properties of the frozen egg yolk liquid with the addition of no protein was characterized using the frequency scanning mode of the rheometer. As illustrated in Figure 2b,c, G′ denotes the solid or elastic nature of the material, while G″ represents its liquid or viscous nature. A comparison of the frozen and thawed samples reveals that the G′ and G″ values of frozen egg yolk liquid with added LDL and HDL were significantly higher than those of the control group. This observation indicates that the increased concentrations of HDL and LDL enhance the interactions between yolk proteins, providing structural support and promoting yolk agglomeration, which in turn leads to an increase in both the elastic and viscous moduli of the yolks [18].

Conversely, the G′ and G″ values of the frozen yolk solution increased with higher levels of LDL and HDL, indicating that the interaction among yolk proteins intensifies as LDL and HDL content rises, thereby exacerbating the aggregation behavior of the proteins during the freezing process. This phenomenon aligns with the findings from the steady-state shear model and the observed results. A strong linear positive correlation was identified between the concentrations of LDL and HDL in the system and the gel strength. A similar trend was reported by Primacella et al. [26], who found that the apparent viscosity of the gel yolk increased with elevated LDL and HDL levels. Thus, the content of LDL and HDL directly influences the strength of the gel following the freezing and thawing of the yolk liquid.

#### 3.2.3. Tangent Loss Analysis

The ratio of G′ to G″, denoted as tanδ, serves as an indicator of the solid or liquid nature of a sample. Specifically, when tanδ > 1, it suggests that the sample exhibits liquid characteristics, whereas tanδ < 1 indicates solid characteristics [25]. As illustrated in Figure 2d, at lower frequencies, the tanδ values of the control samples exceeded 1, demonstrating liquid properties and signifying that the egg yolk liquid had not yet transitioned into a gel. Fei et al. [8] reported that egg yolk proteins aggregate upon freezing to form a physical gel, a process that is more complex than the gel formation driven solely by intermolecular interactions. In contrast, the tanδ values of the samples with added LDL and HDL fell below 1, indicating the presence of solids and confirming that the yolk liquid had indeed formed a gel at this stage. These findings are consistent with previous apparent viscosity measurements. This phenomenon suggests an increased aggregation and gelation of the yolk components during freezing and storage when protein is added. As the amount of added protein increased, the tan δ value decreased, indicating a gradual increase in the content of agglomerates within the frozen egg yolk liquid system.

### 3.3. Moisture Distribution (LF-NMR) Analysis

The water distribution of frozen egg yolks, melted with added LDL versus HDL, was determined using nuclear magnetic resonance (LF-NMR). As illustrated in Figure 3a,b, the relaxation curve of egg yolk exhibits three peaks, each representing a distinct type of water [15]. Previous studies have indicated that the relaxation time reflects the mobility of water in various states. Specifically, it has been demonstrated that a shorter relaxation time (T2) correlates with an enhanced capacity of the sample to bind water. Accordingly, T2b (0 m s to 10 m s), denoted as F1, represents bound water; T21 (10 m s to 100 m s), referred to as F2, signifies fixed water; and T22 (100 m s to 1000 m s), identified as F3, corresponds to free water. The largest distribution area of T21 suggests that non-fragmentable water (freezable water) constitutes the primary state of yolk moisture. This conclusion aligns with findings by Au et al. [19], who also investigated the moisture distribution in eggs.

Compared to the control group, the number of peaks associated with added LDL and HDL yolks remained unchanged. The percentage of water that was not easily flowable remained the highest. As the content of added protein increased, the positions of the T2b peaks did not exhibit significant changes; however, the T21 and T22 peaks shifted toward higher relaxation times, and the peak areas also varied. The percentage of water in different states was calculated based on the magnitudes of the individual peak areas. It has been suggested that aggregated egg yolk proteins may connect to form a continuous network, which reduces the freedom of water molecules and results in a decreased relaxation time. The frozen egg yolk samples with added LDL and HDL demonstrated greater T21 and T22 values compared to the control group, and the increased relaxation time indicated a rise in water mobility, suggesting that more water solidified into ice crystals. These findings imply that LDL and HDL facilitate the growth of ice crystals [27].

In contrast, the bound water content of freeze–thawed egg yolks supplemented with LDL and HDL was higher than that of the blank control. This increase may be attributed to the ability of LDL and HDL to minimize the loss of surface-bound water during the freezing process, which is associated with protein denaturation and aggregation. Additionally, the fixed water content in the freeze–thawed egg yolks with LDL and HDL was lower than that of the blank control. This reduction may result from the aggregation of proteins in the egg yolks, facilitated by surface hydrophobic interactions or other forces during freezing and storage. This aggregation forms a mesh structure that tightly binds some of the water, preventing it from flowing freely within the protein matrix. Consequently, this results in the formation of a water–protein–water polymer, transforming some of the fixed water into bound water. As a result, there is an increase in the bound water content and a corresponding decrease in the fixed water content [12]. The decrease in the fixed and free water content following the freezing and thawing of samples with added LDL and HDL can be explained in two ways. Firstly, the fixed water present in the egg yolk is converted into bound water, thereby reducing the amount of water that is not readily mobile. Secondly, during freezing, ice crystals are formed, and as the protein content increases, the size of these ice crystals gradually enlarges, further disrupting the internal structure of the protein. This disruption results in the loss of water from the yolk, particularly free water, which is the least tightly bound [28]. It is hypothesized that the presence of LDL and HDL during freezing increases the bound water content while decreasing the fixed water content. This shift leads to a higher concentration of yolk solids, exacerbating changes in the internal environment of the yolks and resulting in alterations to the protein structure. Consequently, these changes contribute to the phenomenon of yolk gelatinization. Therefore, the increase in bound water, along with the decrease in fixed and free water in egg yolk, is a key factor in the promotion of yolk gelatinization by HDL and LDL.

### 3.4. Raman Spectrum Analysis

The secondary structure of proteins is primarily maintained by non-covalent interactions. When the environment changes, protein molecules undergo conformational rearrangements to attain the lowest energy state, thereby maintaining relative stability. In this study, Raman spectroscopy was employed to analyze the effects of low-density lipoprotein (LDL) and high-density lipoprotein (HDL) on the frozen gelatinization of egg yolks supplemented with LDL and HDL. The alterations in the secondary structure of proteins were examined to elucidate the mechanisms by which LDL and HDL influence the gelation of egg yolks.

The characteristic peaks of the amide I band (1600–1700 cm^−1^) in the Raman spectra are attributed to the amide bond stretching vibration of the peptide bond. The vibration frequency is influenced by the nature of the hydrogen bonding between C=O and N-H, allowing the combination of these factors to reflect the specific secondary structure of the peptide or protein. Notably, the Raman spectral peaks did not exhibit significant changes with the increasing LDL and HDL content in the egg yolk system. A prominent peak at 1520 ± 5 cm^−1^ is observed, corresponding to the C=C stretching vibration of the alkyl chain of egg yolk lipoproteins [29]. In Figure 4, the significant decrease in peak intensity observed upon the addition of LDL and HDL content may be attributed to the covalent binding of the water-OH group to the C=C group in the yolk protein, which weakens the C=C vibration. This interaction likely causes a transition in the protein’s secondary structure from a disordered to an ordered state. Additionally, the formation of ice crystals during freezing may disrupt the conformation of the yolk protein, leading to alterations in its secondary structure. As a result, intermolecular forces intensify, promoting the aggregation of yolk protein. Furthermore, freezing enhances the hydrophobic interactions among yolk proteins and facilitates the formation of intermolecular disulfide bonds, thereby increasing the forces between yolk proteins and contributing to the gelatinization of the yolk liquid system [30].

### 3.5. Fluorescence Spectroscopy Analysis

The endogenous fluorescence of proteins, primarily arising from aromatic amino acids, is widely employed to monitor the conformation of protein tertiary structures. When excited at 280 nm, the fluorescence spectrum is predominantly attributed to the emission from tyrosine (Tyr) and tryptophan residues. Figure 5a illustrates the changes in fluorescence intensity and the maximum emission wavelength (λmax) of frozen–thawed egg yolk proteins as the low-density lipoprotein (LDL) content increases relative to high-density lipoprotein (HDL) during freezing. The λmax of egg yolk samples with added LDL compared to HDL exhibited an increase relative to the control. The λmax is indicative of the positioning of the Tyr residue within the protein molecule. A conformational change in the protein exposes the tryptophan residue, resulting in a redshift in λmax (higher wavelengths) [30]. The extent of this redshift correlates with the degree of variation in protein conformation; thus, a greater redshift indicates a more significant difference in protein structure.

The fluorescence intensity of freeze–thawed egg yolks gradually decreased as the levels of LDL and HDL were increased, suggesting that increasing the levels of LDL and HDL exacerbates the reduction in tryptophan and tyrosine polarity by freezing, and that lipoprotein polymerization during freezing is enhanced, leading to further alterations in tertiary structure [31]. The above results indicate that the addition of LDL and HDL exacerbated freeze-induced protein denaturation, which further altered the tertiary conformation of the egg yolk protein.

### 3.6. Hydrophobicity Analysis

The exposure of a protein’s hydrophobic structure determines its hydrophobicity, with hydrophobic interactions playing a crucial role in the stability, conformation, and functional properties of proteins, thereby influencing their gelation characteristics [2]. Protein surface hydrophobicity, which serves as an indicator of the content of surface hydrophobic groups when proteins interact with their environment, has a more significant impact on function than total hydrophobicity. As illustrated in Figure 5b, the surface hydrophobicity of frozen–thawed egg yolk samples increased with higher levels of LDL and HDL, showing significant elevation compared to the control frozen–thawed egg yolk samples. The hydrophobicity of proteins tends to increase during freezing, and the presence of the elevated HDL and LDL content exacerbates this effect. This phenomenon may be attributed to the role of LDL and HDL in facilitating freezing, which leads to the further unfolding of proteins and the exposure of hydrophobic aliphatic and aromatic amino acids [32]. The increase in surface hydrophobicity may result from the disruption of the protein’s original molecular structure, leading to the exposure of hydrophobic groups that were initially buried within the molecule. This effect may be exacerbated by the addition of low-density lipoprotein (LDL) and high-density lipoprotein (HDL). The unfolding of the protein’s hydrophobic structure contributes to the formation of a denser yolk gel network. Freezing treatment reduces the polarity of the tryptophan and tyrosine microenvironments, thereby enhancing lipoprotein polymerization through hydrophobic interactions. Consequently, increasing levels of LDL and HDL promote gelation within the yolk system, resulting in a higher degree of polymerization [33]. The surface hydrophobicity of the HDL-added group is observed to be slightly greater than that of the LDL-added group. This finding suggests that HDL exposes more hydrophobic groups within the frozen egg yolk system compared to LDL, and the unfolding of the hydrophobic structure results in a higher degree of protein aggregation.

### 3.7. l* Diffusion Wave Spectrometer Analysis

Photon free range *l** represents the average distance that light travels through the system per unit time. As illustrated in Figure 6, it is evident that with increasing additions of LDL and HDL, the *l** value gradually decreases, with the HDL group exhibiting a lower *l** value than the LDL group. A smaller *l** value indicates that bright light encounters greater difficulty in passing through the yolk liquid system, which correlates with an increased intensity of gel formation. This observation suggests that higher concentrations of LDL and HDL facilitate the freezing gelation of egg yolk liquid, with HDL demonstrating a more pronounced effect on gelation. These findings align with the results obtained from rheological experiments and observations of egg yolk during freezing and thawing.

### 3.8. Fluorescence Inverted Microscope Analysis

Micrographs of frozen and thawed egg yolks were examined using a fluorescence inverted microscope, with samples stained for both proteins and lipids, as illustrated in the figures. In Figure 7, proteins are depicted in green and lipids in red. The results indicate that in the frozen–thawed egg yolk samples supplemented with LDL and HDL, both lipids and proteins exhibited irregular shapes and formed larger clusters compared to the control samples, with the frozen–thawed egg yolk particles showing an uneven distribution. As the concentrations of LDL and HDL increased, the aggregated clusters became larger and heavier. This phenomenon may be attributed to the elevated levels of LDL and HDL, which could exacerbate the formation of micelles and granular structures within the freeze–thawed egg yolk proteins, alongside the accumulation of polar and non-polar lipid molecules [34]. These observations suggest that the gelatinization occurring during the freeze–thawing process of the egg yolk liquid may result from the aggregation of lipoproteins and lipids, such as LDL and HDL. Furthermore, the particle aggregation in the HDL group was more pronounced, leading to a more evident gel formation in the egg yolk. This implies that HDL may play a more significant role in the gel formation associated with the freezing of the egg yolk liquid, which is further supported by the increased apparent viscosity observed in the HDL group, as reflected in the rheological results and accompanying phenomena.

## 4. Discussion

In our experimental results, we observed that the freezing gelation of the egg yolk liquid became more pronounced with an increased ratio of LDL to HDL. This phenomenon may be attributed to the fact that the aggregation of egg yolks is influenced by variations in forces during the freezing process, and the gelatinization of egg yolks is associated with the breakdown of lipoproteins and the subsequent interactions among protein molecules [35]. An increased LDL to HDL ratio enhances intermolecular forces. Following the freeze–thaw treatment, the hydrophobic and electrostatic interactions between egg yolk proteins were intensified, leading to the formation of a dense gel structure. Chi et al. [36] proposed that gelation results from non-specific aggregation due to the cross-linking of peptide or phospholipid polar head groups of neighboring lipoprotein particles. During freezing, as water in the solvent transitions to ice, contacts among particles are established and strengthened, particularly as the concentration of particles increases with a higher LDL to HDL ratio, which enhances protein aggregation. Additionally, we found that increasing the percentage of HDL had a more significant impact on the gelatinization of the egg yolk liquid compared to the LDL group. The reason for this may be the greater involvement of egg yolk HDL in the freeze–thaw gelatinization during cryopreservation. Compared with LDL, HDL has a lower lipid content and a stronger freeze–thaw process, which protects lipoproteins from freeze–thaw denaturation, thus allowing more lipoprotein aggregation to occur [37]. In this study, we focused exclusively on LDL and HDL, the two primary lipoproteins present in egg yolk, to investigate their mechanisms of influence on the cryogelation of the liquid egg yolk. However, the exploration of the mechanisms underlying egg yolk liquid gelation remains underdeveloped, and the potential interactions between proteins and lipids that may also affect cryogelation were not analyzed. Future research should consider modifying methods to isolate LDL and HDL with greater purity while preserving their natural physiological conditions and properties. Additionally, further experiments are warranted to examine the effects of other constituents in egg yolk, such as phospholipids and sterols, on the gelation of liquid egg yolk.

## 5. Conclusions

The present study confirms that both LDL and HDL play significant roles in the cryogelation process of egg yolk. It demonstrates that as LDL aggregates, HDL interacts with these LDL aggregates, contributing to gel formation. Increasing the levels of LDL and HDL intensifies the gelatinization of the liquid egg yolk, with the enhancement caused by HDL being particularly pronounced. This indicates that HDL may exert a greater influence on the frozen gelation of egg yolks. Elevated levels of both LDL and HDL resulted in an increase in bound water and a decrease in fixed free water during egg yolk gelation, thereby exacerbating the gelation process. Additionally, the increased content of LDL and HDL was found to contribute to changes in protein conformation due to ice crystal formation, leading to the further unfolding of hydrophobic groups in egg yolks during freezing. Observations of the microstructure of frozen and thawed egg yolks reveal that a higher LDL and HDL content correlates with the more pronounced aggregation of proteins and lipids, suggesting that the gelation of the liquid egg yolk may result from the aggregation of substances such as LDL and HDL.

## Figures and Tables

**Figure 1 foods-14-00522-f001:**
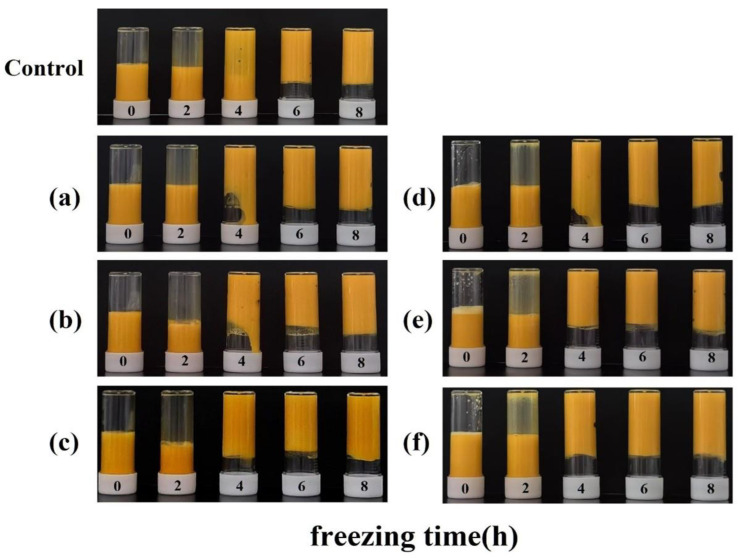
Observations of the phenomena of egg yolk liquid supplemented with 3%, 6%, and 9% LDL and HDL after freezing and thawing at −37 °C; (**a**–**c**) LDL from 3 to 9%; (**d**–**f**) HDL from 3 to 9%.

**Figure 2 foods-14-00522-f002:**
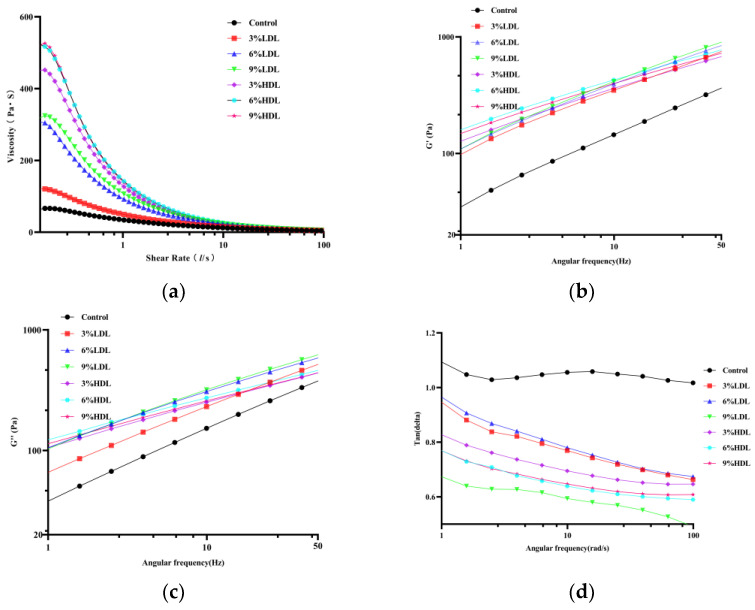
Rheological properties of frozen–thawed egg yolk mixtures: (**a**) apparent viscosity of the egg yolk mixture; (**b**) modulus of elasticity(G′) of frozen–thawed egg yolk mixtures; (**c**) loss modulus(G″) of frozen–thawed egg yolk mixtures; and (**d**) tangent loss (tanδ) of frozen–thawed egg yolk mixtures.

**Figure 3 foods-14-00522-f003:**
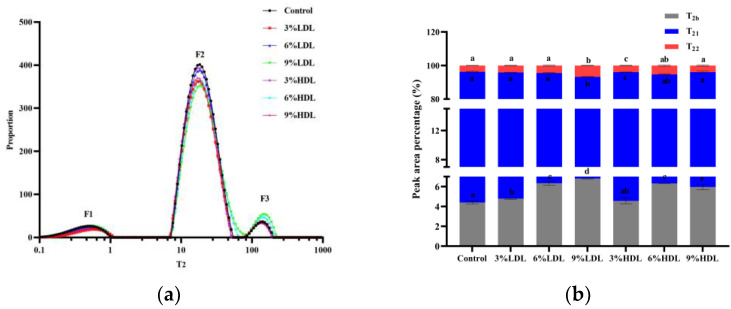
Moisture distribution images of freeze–thawed egg yolk mixtures: (**a**) changes in the T_2_ representative distribution of freeze–thawed egg yolk mixtures; (**b**) peak area of freeze–thawed egg yolk mixtures. Values with different letters are significantly different (*p < 0.05*).

**Figure 4 foods-14-00522-f004:**
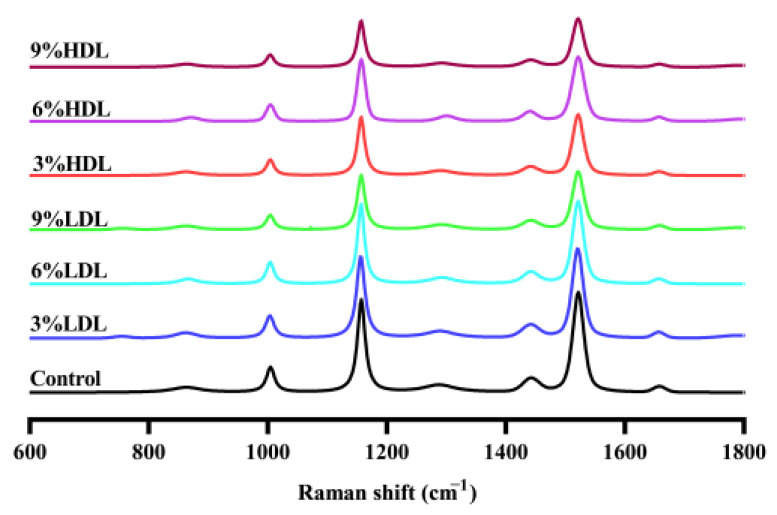
Raman spectra of freeze–thawed egg yolk mixtures.

**Figure 5 foods-14-00522-f005:**
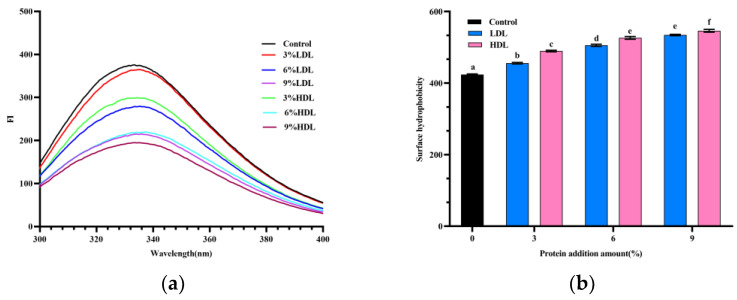
Protein tertiary structure of freeze–thawed egg yolk mixtures: (**a**) fluorescence spectra of freeze–thawed egg yolk mixtures; (**b**) hydrophobicity of freeze–thawed egg yolk mixtures. Values with different letters are significantly different (*p < 0.05*).

**Figure 6 foods-14-00522-f006:**
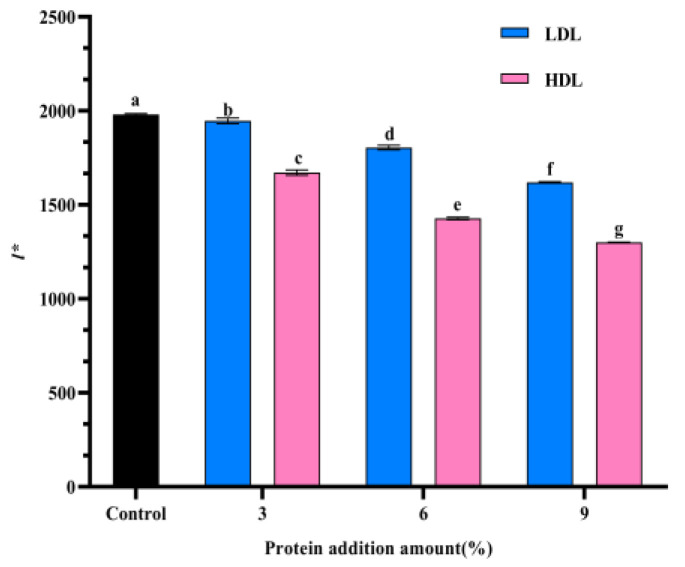
Optical free range (*l**) of freeze–thawed egg yolk mixtures. Values with different letters are significantly different (*p < 0.05*).

**Figure 7 foods-14-00522-f007:**
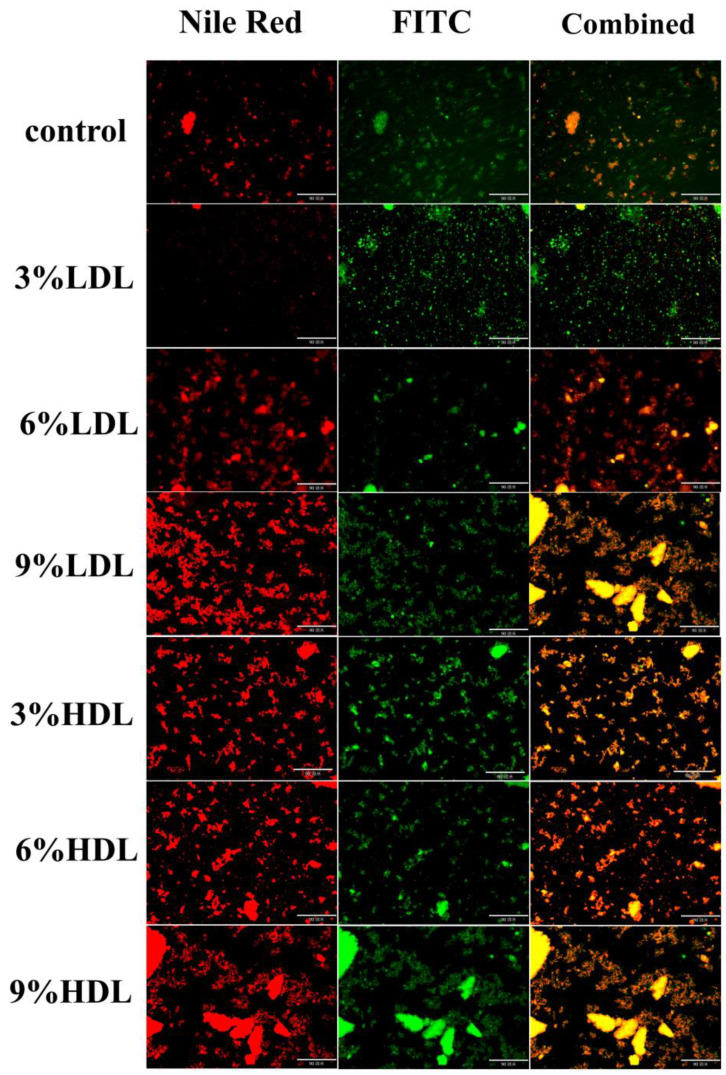
Fluorescence staining micrographs of freeze–thawed egg yolk mixtures. The unite of length is 90 μm.

## Data Availability

Data will be made available on request.

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
