# Peer review of "Effect of Low-Density Lipoprotein (LDL) and High-Density Lipoprotein (HDL) on Frozen Gelation of Egg Yolk"

_foods, 2025, doi:10.3390/foods14030522_

Round 1

Reviewer 1 Report

Comments and Suggestions for Authors

The paper (Effect of low-density lipoprotein (LDL) and high-density lipoprotein (HDL) on frozen gelation of egg yolk) is well written, and the study is considered novel. In this study, the effect of the addition of LDL and HDL isolated from egg yolk to egg yolk at different ratios (3,6 and 9%) on the gelling property of egg yolk was investigated.  The experimental design and analyses were found to be compatible and appropriate to the topic of the article. The manuscript needs improvement in the suggested following issues.

Abstract:

-Line 19: G' and G'' values, with italicized symbol

-Line 20 …..with the addition of ….., please add to make more clear

-Line 28: Indicate above a numerical value?

-Line 32: Which is more preferable for these purposes, LDL or D-HDL?

-Line 94: aw

-Line 135: specify the characteristics of the filter (diameter, etc.).

-Line 140: “The supernatant was then collected” It is like repeating the end of the previous sentence.

-Equation 1:please use this symbol instead of ε

-Line 186: 2.5.2 Viscoelasticity          

- Figures 2b and 2c, please use angular frequency on the x-axes and the values on the y-axes should be in logarithmic scale.

-Line 302: It should also be discussed that there is no difference between 6% and 9% and that the increase in concentration of HDL does not affect viscosity.

-Line 303: Do not make G' to G'' and tanδ discussions under the heading of viscosity.

-Harmonies the discussed parts and the modulus, tan delta parameters with the order in the figure. tand delta is discussed first but the order is 2d.

-Line 333: “Primacella et al. [17]” HDL is not used in the relevant article, please change your discussion or citation.

Comments on the Quality of English Language

sufficient and appropriate.

Reviewer 2 Report

Comments and Suggestions for Authors

General comments

The manuscript entitled "Effect of low-density lipoprotein (LDL) and high-density lipoprotein (HDL) on frozen gelation of egg yolk" presents a very interesting investigation about the effect of the quantity of these two major egg yolk proteins on the frozen gelation phenomenon. The text is well-written, and the analyses performed are very enlightening. However, some points need to be clarified and/or addressed before potential publication.

Specific comments

1. In Figure 1, differences between the behavior of samples with added HDL and LDL can be observed. This difference is more pronounced at 2 h and 4 h of freezing. However, no differentiation between the behavior of the two types of samples is presented in the text. The authors should discuss this point in the manuscript.

2. The authors conducted the rheological study only for a freezing time of 4 hours. Performing this study with different freezing times would have provided very interesting information about gelation under freezing. This would also contribute to understanding the behavior observed in Figure 1.

In this context, why were the flow curve and oscillatory test not performed for different freezing times?

3. Lines (180 – 184)

The authors chose the Power Law to suggest that they would fit the parameters of the flow curve.

3.1. Why was this equation chosen? Were other rheological models studied?

3.2. Correct equation 1 to the proper format.

3.3. Where are the parameter values for the rheological model (consistency coefficient and flow behavior index) as well as the determination coefficient for the model presented?

4. Change the “viscosity” for “ apparent viscosity” in all the text

5. Topic “3.2 Rheological Analysis”

Firstly, the results regarding tan𝛿 are presented and discussed, followed by the results on loss moduli and elasticity moduli. The authors should reverse this order in the text.

6. From Figure 5, it can be observed that the addition of HDL has a greater effect on increasing hydrophobicity than the addition of LDL for the same increase in lipoprotein concentration. The authors must to discuss this difference in the text.
